# Transcriptional Regulator DasR Represses Daptomycin Production through Both Direct and Cascade Mechanisms in *Streptomyces roseosporus*

**DOI:** 10.3390/antibiotics11081065

**Published:** 2022-08-05

**Authors:** Qiong Chen, Jianya Zhu, Xingwang Li, Ying Wen

**Affiliations:** 1State Key Laboratory of Agrobiotechnology and College of Biological Sciences, China Agricultural University, Beijing 100193, China; 2Institute of Fisheries Research, Beijing Academy of Agriculture and Forestry Sciences, Beijing 100068, China

**Keywords:** *Streptomyces roseosporus*, daptomycin, morphological development, DasR, AdpA

## Abstract

Daptomycin, produced by *Streptomyces roseosporus*, is a clinically important cyclic lipopeptide antibiotic used for the treatment of human infections caused by drug-resistant Gram-positive pathogens. In contrast to most *Streptomyces* antibiotic biosynthetic gene clusters (BGCs), daptomycin BGC has no cluster-situated regulator (CSR) genes. DasR, a GntR-family transcriptional regulator (TR) widely present in the genus, was shown to regulate antibiotic production in model species *S. coelicolor* by binding to promoter regions of CSR genes. New findings reported here reveal that DasR pleiotropically regulates production of daptomycin and reddish pigment, and morphological development in *S. roseosporus. dasR* deletion enhanced daptomycin production and morphological development, but reduced pigment production. DasR inhibited daptomycin production by directly repressing *dpt* structural genes and global regulatory gene *adpA* (whose product AdpA protein activates daptomycin production and morphological development). DasR-protected regions on *dptEp* and *adpAp* contained a 16 nt sequence similar to the consensus DasR-binding site *dre* in *S. coelicolor*. AdpA was shown to target *dpt* structural genes and *dptR2* (which encodes a DeoR-family TR required for daptomycin production). A 10 nt sequence similar to the consensus AdpA-binding site was found on target promoter regions *dptAp* and *dptR2p*. This is the first demonstration that DasR regulates antibiotic production both directly and through a cascade mechanism. The findings expand our limited knowledge of the regulatory network underlying daptomycin production, and will facilitate methods for construction of daptomycin overproducers.

## 1. Introduction

Gram-positive *Streptomyces* species are abundant in soil and decaying vegetation. Their morphological development (involving formation of substrate hyphae, aerial hyphae, and chains of spores) is typically associated with the synthesis of secondary metabolites, including antibiotics that display well-documented antibacterial, antiviral, antifungal, anthelmintic, anticancer, and immunosuppressive activities. Antibiotic biosynthetic genes are usually clustered, with one or more cluster-situated regulator (CSR) genes located within the cluster. Antibiotic production is tightly controlled by complex regulatory networks based on CSRs and various higher-level pleiotropic/global regulators that respond to environmental and physiological changes [1,2,3].

*Streptomyces roseosporus* produces daptomycin, a clinically important cyclic lipopeptide antibiotic that strongly inhibits multidrug-resistant Gram-positive pathogens, including methicillin-resistant *Staphylococcus aureus* (MRSA), through a distinctive action mechanism [4]. Daptomycin is used as a “last resort” treatment for right-sided infective endocarditis and complex skin and soft tissue infections [5]. It contains 13 amino acids to form a 10-membered cyclic peptide and a 3-amino-acid tail with a straight-chain decanoic acid moiety [6]. Daptomycin is biosynthesized through a nonribosomal peptide synthase (NRPS) pathway [7], and its biosynthetic cluster *dpt* consists of 12 structural genes. Among these, *dptA, dptBC*, and *dptD* encode three subunits of NRPS; *dptE* and *dptF* are involved in activation of decanoic acid; and *dptG*, -*H*, -*I*, -*J*, -*M*, -*N*, and -*P* are responsible for precursor supply, resistance, or transport [8,9]. Three regulatory genes (*dptR1*, *dptR2*, *dptR3*) are located adjacent to the *dpt* structural genes. In view of the clinical and commercial importance of daptomycin, many studies have addressed the regulatory mechanism of its biosynthesis. Our group and YQ Li’s group observed that DptR1 did not affect daptomycin production [10,11], whereas GH Yu’s group reported that daptomycin production was repressed by both deletion and overexpression of *dptR1* [12]. The reason for these seemingly contradictory findings is unclear. DptR2 is required for daptomycin production, but does not regulate the expression of the *dpt* cluster [13]. We found that DptR3 promotes daptomycin production by activating transcription of *dpt* structural genes in an indirect manner [11]. These findings suggest that the *dpt* cluster does not contain CSR genes, in contrast to other typical antibiotic biosynthetic gene clusters (BGCs) of *Streptomyces*. YQ Li’s group demonstrated that transcriptional regulators (TRs) AtrA [14], DepR1 [10], DepR2 [15], and PhaR [16] regulate daptomycin production by binding directly to *dptEp*, the promoter for core operon *dptE-dptF-dptA-dptBC-dptD-dptG-dptH* of the *dpt* cluster. They also reported that two global regulators, AdpA and PhoP, bind competitively to *atrAp*, activate *atrA* expression, and thereby promote daptomycin production [17]. WblA (WhiB-family TR) inhibits daptomycin production [18], and the cyclic AMP receptor protein Crp has a positive effect on daptomycin production [19], but whether their regulatory effect is direct or indirect is unclear. We recently demonstrated [20] that BldD, the master repressor of *Streptomyces* development, also activates daptomycin production directly and through a cascade mechanism based on binding to *dptEp*, *dptR3p*, *adpAp*, and *afsRp*. Our knowledge of the regulatory network underlying daptomycin production remains quite limited and fragmentary despite the above findings, presenting an obstacle to the rational design of daptomycin high-yielding strains through genetic manipulation.

DasR, a GntR-family TR, was initially identified in model species *S. coelicolor* as a global regulator involved in control of GlcNAc/chitin catabolism [21,22], antibiotic production [23], and morphological development [22]. Consensus DasR-binding site *dre* (DasR-responsive element) is a 16 nt imperfect palindromic sequence 5′-DSWGGWSTVVDCMHBN-3′ (D = A/G/T; S = C/G; W = A/T; V = A/C/G; M = A/C; H = A/C/T; B = G/C/T; N = A/G/C/T) [3,21]. DasR represses actinorhodin (Act) and undecylprodigiosin (Red) production by binding to *dre* sites upstream of cluster-situated activator genes *actII-ORF4* and *redZ*, respectively [23]. DasR is also essential for *S. coelicolor* development; *dasR* deletion results in “bald” phenotype [22]. In *S. cinnamonensis*, DasR acts as an activator of monensin production by binding to promoter regions of CSR and structural genes of *mon* cluster [24]. DasR homologs are present in actinomycetes other than *Streptomyces*, including the erythromycin producer *Saccharopolyspora erythraea*. DasR in this species directly activates reddish pigment production by binding to *dre* site upstream of *rpp* cluster, and indirectly promotes erythromycin production. *dasR* deletion in *S. erythraea* causes delayed development, but does not abolish formation of aerial hyphae and spores as it does in *S. coelicolor* [25]. The above findings reflect the diverse roles of DasR in secondary metabolism and development, and the complex regulatory mechanisms involved. To date, DasR functions have not been investigated in *S. roseosporus*.

Here, we describe the characterization of DasR in *S. roseosporus* as a dual repressor/activator, i.e., repressor of daptomycin production and morphological development, and activator of pigment production. Its inhibitory effect on daptomycin production is mediated by both direct and cascade mechanisms, based on binding to *dptEp* and *adpAp*. Our findings demonstrate the essential role of DasR in control of *Streptomyces* antibiotic production.

## 2. Results

### 2.1. DasR Inhibits Daptomycin Production and Morphological Development, but Promotes Pigment Production

*S. roseosporus dasR* gene contains 765 nucleotides (nt) and encodes a 254-amino-acid protein, DasR. DasR is a highly conserved protein, as revealed by sequence alignment analysis; it has 89.8, 92.1, 98.8, and 93.7% amino acid identities with its homologs in *S. coelicolor*, *S. avermitilis*, *S. griseus*, and *S. venezuelae*, respectively, reflecting its important functional role in the genus.

To investigate the role of DasR in *S. roseosporus*, the *dasR*-deletion mutant DdasR was constructed by homologous recombination (Appendix A). Daptomycin production by DdasR grown in fermentation medium for 10 days was 26% higher than that of the wild-type (WT) strain NRRL11379 [26] (Figure 1A). Complementation of DdasR (strain CdasR) restored daptomycin production to WT level. Enhancement of *dasR* expression in WT (strain OdasR) reduced daptomycin production by 25%. For plasmid control strains WT/pKC1139 and WT/pSET152, daptomycin production was close to that of WT (Figure 1A). Time course measurements of growth and daptomycin production showed that biomass (dry cell weight) values of DdasR and OdasR were similar to that of WT (Figure 1B), whereas daptomycin production was upregulated by *dasR* deletion and downregulated by *dasR* overexpression (Figure 1C). These findings indicate that DasR acts as a repressor of daptomycin production and has no effect on cell growth.

WT, DdasR, CdasR, and OdasR were grown on DA1 plates for sporulation in order to assess the effect of DasR on morphological development. Spore formation occurred earlier for DdasR, whereas development of CdasR and OdasR did not differ notably from that of WT (Figure 1D), indicating that DasR is a repressor of *S. roseosporus* development.

*S. roseosporus* produces a reddish pigment (a secondary metabolite). Pigment production was much lower for DdasR than for WT, but did not differ notably for CdasR or OdasR (Figure 1D), indicating that DasR is an activator of pigment production.

### 2.2. DasR Regulates the Transcription of dpt Genes

To investigate the causes of daptomycin overproduction in DdasR, we performed RT-qPCR analysis of RNAs prepared from WT and DdasR cultured in fermentation medium at days 2 (exponential phase), 4 (early stationary phase), and 6 (middle stationary phase). Transcription levels of structural genes *dptA*, *dptBC*, *dptD*, *dptE*, *dptF*, *dptG*, *dptH*, *dptI*, *dptM*, and *dptP* in the *dpt* cluster were higher for DdasR than for WT at one, two, or all three of these time points (Figure 2). These data are consistent with the daptomycin production data. In DdasR, all *dpt* structural genes were upregulated on day 2, suggesting that DasR represses transcription of these genes mainly during the early fermentation stage.

There are three regulatory genes (*dptR1*, *dptR2*, *dptR3*) adjacent to the *dpt* cluster. Because we found that DptR1 was not involved in control of daptomycin production [11], we only assessed transcription levels of *dptR2* and *dptR3* in WT and DdasR using the same RNA preparations. RT-qPCR results revealed that *dptR2* was downregulated, whereas *dptR3* was upregulated in DdasR (Figure 2). DptR2 and DptR3 were previously shown to promote daptomycin production [11,13]. Therefore, the observed change of daptomycin production in DdasR reflects a combined effect of altered transcription levels of *dpt* structural and regulatory genes.

### 2.3. DasR Binds Specifically to the dptE-Promoter Region

To determine whether the *dpt* genes described above are regulated directly by DasR, we performed electrophoretic mobility shift assays (EMSAs) using soluble His_6_-DasR purified from *E. coli* and respective promoter probes. The *dpt* cluster consists of four transcriptional units: *dptP*, *dptM*-*dptN*, *dptE-dptF-dptA-dptBC-dptD-dptG-dptH* (hereafter termed “*dptE* operon”), and *dptI-dptJ* [20,27] (Figure 3A). *dptA*, *dptBC*, and *dptD* are large, contiguous genes that encode subunits of NRPS for biosynthesis of daptomycin skeleton. We therefore performed 5′ RACE analysis to test the possibility that *dptA* has its own transcriptional start site (TSS). *dptA* TSS was mapped to G, 453 nt upstream of *dptA* translational start codon (TSC) (Appendix A), indicating that *dptA* has its own promoter for *dptA-dptBC-dptD-dptG-dptH* (hereafter termed “*dptA* operon”). Promoter probes *dptP-M* (containing bidirectional promoters), *dptEp*, *dptAp*, *dptIp*, *dptR2p*, and *dptR3p* were designed and used for EMSAs (Figure 3A). TRs generally do not bind to open reading frame (ORF) of genes. We therefore used nonspecific probe *hrdB* within *hrdB* ORF as negative control. EMSA results indicated that His_6_-DasR formed complexes with probe *dptEp*, but not with *dptP-M*, *dptAp*, *dptIp*, *dptR2p*, *dptR3p*, or *hrdB* (Figure 3B). Binding specificity was evaluated by competition assays using ~100-fold excess unlabeled specific probe *dptEp* (lane S) (which abolished delayed bands) and *hrdB* (lane N) (which did not). Our findings indicate that DasR directly represses *dptE* operon, whereas it represses *dptI*, *dptM*, *dptP*, and *dptR3*, and activates *dptR2*, in an indirect manner.

DNase I footprinting assays were performed to determine the precise DasR-binding site on *dptEp*, and to clarify the mechanism whereby DasR regulates target *dptE* operon. DasR protected a 25 nt region containing a 16 nt sequence (5’-AGTGGTTTGGTCCGCC-3’) (Figure 4A), similar to the conserved DasR-binding site *dre* (5’-DSWGGWSTVVDCMHBN-3’) (D = A/G/T; S = C/G; W = A/T; V = A/C/G; M = A/C; H = A/C/T; B = G/C/T; N = A/G/C/T) in *S. coelicolor* [3,21], suggesting that the DNA-binding property of DasR is conserved in the genus.

We previously mapped the TSS of *dptE* to A, 267 nt upstream of *dptE* TSC [20], and accordingly predicted −10 and −35 promoter elements (Figure 4B). The 16 nt DasR-binding site was found to extend from positions +72 to +87, relative to *dptE* TSS (Figure 4B). DasR presumably represses *dptE* operon transcription by blocking transcriptional extension of RNA polymerase.

### 2.4. DasR Directly Represses adpA Involved in Daptomycin Production and Morphological Development

Additional DasR target genes involved in daptomycin production were investigated by using *dre* sequence with MAST/MEME program (http://meme-suite.org, accessed on 29 June 2022) to scan the *S. roseosporus* genome. Global regulatory gene *adpA* was identified as a putative DasR target. AdpA protein distributes widely among *Streptomyces* species, and controls antibiotic production and morphological development [1]. The accuracy of bioinformatic prediction was tested by performing EMSAs using His_6_-DasR and promoter probe *adpAp*. DasR bound specifically to *adpAp* (Figure 5A), indicating that it targets *adpA. adpA* transcription level was enhanced at three time points during growth of DdasR in fermentation medium (Figure 5B), indicating that DasR acts as a repressor of *adpA*.

We reported previously that *adpA* TSS is localized at C, 263 nt upstream of *adpA* TSC [20] DNase I footprinting assays revealed that the 40 nt protected site of DasR on *adpAp* extends from positions +14 to +53 nt relative to *adpA* TSS, and contains a 16 nt *dre*-like sequence 5′-GAAGGGCACTTCCCTG-3′ (Figure 5C,D). This sequence is located downstream of *adpA* TSS (Figure 5D), suggesting that DasR represses *adpA* transcription through a mechanism similar to that of *dptE* operon repression.

The 2015 study by YQ Li’s group indicated that AdpA promotes daptomycin production and morphological development in *S. roseosporus* [14]. That study involved construction of an *adpA*-deletion mutant, based on WT strain SW0702, that abolished daptomycin production, but did not involve construction of an *adpA* overexpression strain. We further investigated the role of AdpA in *S. roseosporus* by constructing *adpA*-deletion mutant DadpA (Appendix A), complemented strain CadpA, and overexpression strain OadpA based on WT strain NRRL11379. Quantitative analysis of cultures following 10-day fermentation revealed that, relative to WT, daptomycin production was reduced by 33% in DadpA, and increased by 49% in OadpA (Figure 6A). Daptomycin production by CadpA was close to that of WT, demonstrating that reduced production in DadpA was due solely to *adpA* deletion. Examination of WT, DadpA, and OadpA growth and fermentation curves revealed that *adpA* deletion and overexpression did not notably affect cell growth (Figure 6B), but reduced and increased daptomycin production, respectively (Figure 6C). The above findings clearly indicate that AdpA positively regulates daptomycin production, regardless of the reduced production observed for DadpA. Enhanced *adpA* expression in *dasR*-deletion mutant DdasR thus contributes to increased daptomycin production.

The role of AdpA in *S. roseosporus* morphological phenotype was investigated by streaking WT, DadpA, CadpA, and OadpA strains on DA1 plates. DadpA displayed “bald” phenotype, i.e., grew in substrate mycelia (Figure 6D), consistent with the 2015 study by YQ Li’s group [14]. CadpA and OadpA were phenotypically similar to WT (Figure 6D). These findings indicate that normal development requires the presence of AdpA. Pigment production showed no notable changes in any of the four strains (Figure 6D), suggesting that AdpA is not involved in this process.

### 2.5. AdpA Directly Regulates Expression of dptA Operon and dtpR2

The 2015 study by YQ Li’s group identified *atrA*, which encodes TetR-family TR AtrA that directly activates expression of *dptE* operon, as an AdpA target [14]. To determine whether AdpA also directly regulates *dpt* genes, we performed a series of EMSAs using soluble His_6_-AdpA and *dpt* promoter probes. His_6_-AdpA clearly retarded probes *dptAp* and *dptR2p*, but did not bind to *dptP-M*, *dptEp*, *dptIp*, *dptR3p*, or control probe *hrdB* (Figure 7A), indicating that *dptA* operon and *dptR2* are novel AdpA targets. Effects of AdpA on expression of target *dpt* genes were assessed by RT-qPCR. In DadpA grown in fermentation medium, transcription levels of *dptA*, *dptBC*, *dptD*, *dptG*, and *dptH* within *dptA* operon were reduced at two (days 2 and 4) or three time points (days 2, 4, and 6), and that of *dptR2* was increased at days 2 and 4 (Figure 7B), indicating that AdpA functions as an activator of *dptA* operon but as a repressor of *dptR2*. AdpA was shown to be an activator of *atrA*, and *adpA* deletion resulted in decreased *atrA* expression [14]. Therefore, reduced daptomycin production in DadpA is a collective result of altered expression of AdpA-targeted *atrA* and *dpt* genes.

We determined precise AdpA-binding sites on target promoters *dptAp* and *dptR2p* by DNase I footprinting assays. AdpA protected a 35 nt region extending from +26 to +60 nt relative to *dptA* TSS and containing a 10 nt sequence (5′-TGGCACGCCA-3′) (Figure 8A), similar to the consensus AdpA-binding site 5′-TGGCSNGWWY-3′ (S = C/G; N = A/G/C/T; W = A/T; Y = C/T) [28]. Binding of a transcriptional activator to a site downstream of the TSS is uncommon; however, BldD [20] and DepR1 [10] were shown to activate *dptE* by binding to sites downstream of *dptE* TSS. The regulatory mechanism underlying such transcriptional activation remains to be elucidated.

The *dptR2* TSS was identified by 5′ RACE and mapped to A, 65 nt upstream of *dptR2* TSC (Appendix A). A 38 nt AdpA protected region was detected on *dptR2p*, located −187 to −150 nt relative to *dptR2* TSS (Figure 8B). A 10 nt sequence (5′-TGGCCGATTT-3′) similar to the consensus AdpA-binding site was also found in the AdpA-protected region (Figure 8B). This is analogous to our previous finding that binding sites of AvaR2 (TetR-family TR) on *aveRp* (promoter of cluster-situated activator gene *aveR*) are far upstream of *aveR* TSS, and that AvaR2 represses *aveR* [29]. The mechanism underlying transcriptional repression in this case also remains to be elucidated.

## 3. Discussion

The role of DasR in control of antibiotic production has been well characterized in *S. coelicolor* and *S. cinnamonensis*. DasR inhibits Act and Red production through CSR genes [23], and promotes monensin production through both CSR and biosynthetic genes [24]. The present study was focused on the molecular mechanism whereby DasR regulates production of daptomycin, a clinically important antibiotic whose BGC lacks CSRs, and revealed that DasR acts as a repressor in this process. DasR directly represses expression of *dpt* biosynthetic genes and of regulatory gene *adpA*, whose product AdpA controls daptomycin production through activation of *atrA* [14] and *dptA* operon, but repression of *dptR2*. Thus, DasR regulates daptomycin production both directly and in a cascade manner. This is the first report of such cascade regulatory mechanism of DasR for antibiotic production control in *Streptomyces*. DasR and AdpA are widely present in *Streptomyces* species. Therefore, DasR presumably regulates AdpA in other species besides *S. roseosporus*.

*adpA*, a target of DasR, is a global regulatory gene involved in secondary metabolism and morphological development in *Streptomyces*. It was first identified in *S. griseus* and shown to be directly repressed by A-factor receptor ArpA [30]. Derepression of *adpA* causes activation of secondary metabolic and developmental processes. In *S. coelicolor*, *adpA* is directly repressed by its encoding protein AdpA [31], BldD [32], and pseudo γ-butyrolactone receptor ScbR2 [33], and plays essential roles in Act and Red production and morphological development. In *S. ansochromogenes*, AdpA is required for nikkomycin production and morphological development [34], but inhibits oviedomycin production [35]. In *S. xiamenensis*, AdpA represses development and differentially regulates production of polycyclic tetramate macrolactams and xiamenmycin [36]. In *S. roseosporus*, *adpA* has been demonstrated to be repressed by ArpA [14] and activated by BldD [20]. We found in the present study that *adpA* is also repressed by DasR. Thus, AdpA plays differing roles in secondary metabolism and morphological development in various *Streptomyces* species, and regulation of its expression is complex. The *adpA*-deletion mutant constructed based on *S. roseosporus* SW0702 in the 2015 study by YQ Li’s group abolished daptomycin production [14], whereas the mutant constructed based on strain NRRL11379 [26] in the present study reduced—but did not abolish—daptomycin production. This seeming discrepancy may be due to differences in the parental strains and the growth media used.

*dptEp* drives the expression of the core *dptE* operon, which contains seven daptomycin biosynthetic genes and is the major promoter within the *dpt* cluster. Transcriptional regulation of *dptEp* is highly complex; to date, six TRs have been shown to target this promoter: AtrA [14], DepR1 [10], DepR2 [15], PhaR [16], BldD [20], and DasR (present study). The signaling molecules that DepR1, DepR2, and PhaR sense and respond to are unclear. *atrA* is targeted by AdpA, a key TR in the A-factor signaling pathway; therefore, it is likely that extracellular A-factor-like signals affect *dptE* operon expression via AtrA [14]. BldD’s control of its targets is based on its response to signaling molecule c-di-GMP [37]. The DNA-binding activity of DasR is modulated by multiple signaling molecules that act as allosteric effectors. Binding of DasR to its target DNAs is inhibited by glucosamine-6-phosphate (GlcN-6P) and N-acetylglucosamine-6-phosphate (GlcNAc-6P) catabolized from N-acetylglucosamine (GlcNAc), but enhanced by organic phosphate metabolites from glucose catabolism (e.g., glucose-6-phosphate, glucose-1-phosphate, glycerol-3-phosphate, fructose-1,6-bisphosphate, fructose-6-phosphate) and by inorganic phosphate [22,38]; The metabolic status of a particular cell thus determines the effect of DasR on expression of its targets. In a well-established *S. coelicolor* model, GlcN-6P and GlcNAc-6P (metabolic intermediates of GlcNAc released by cell wall autolysis of substrate mycelium during development) act as DasR allosteric effectors, impair repressive effect of DasR on target CSR genes, and thereby induce antibiotic production [22,23,38]. In view of our findings that DasR represses daptomycin production, and that its target promoters *dptEp* and *adpAp* both contain consensus *dre*-like site, it seems likely that DasR responds to GlcN-6P and GlcNAc-6P in control of daptomycin production in *S. roseosporus*, just as its role in antibiotic production in *S. coelicolor*. The possibility cannot be ruled out, however, that other metabolites (e.g., phosphorylated sugars) act as signaling molecules in DasR-mediated regulation of daptomycin production, depending on culture conditions. Identification of additional environmental or physiological signals that affect daptomycin production will help clarify the complex regulatory mechanisms of this process.

We propose a model of DasR-mediated regulation of daptomycin production in *S. roseosporus* (Figure 9), based on present and previous findings. According to this model, DasR exerts its regulatory effect on daptomycin production through several mechanisms: (i) direct repression of *dptE* operon (which contains *dptE*, *dptF*, *dptA*, *dptBC*, *dptD*, *dptG*, and *dptH*); (ii) indirect repression of other *dpt* structural genes (*dptP*, *dptM*, *dptI*); (iii) direct repression of regulatory gene *adpA*, which activates daptomycin production; (iv) indirect activation of *dptR2* through global regulator AdpA (and perhaps other regulators); (v) indirect repression of *dptR3* through yet-unknown mechanism(s). DasR also regulates morphological development and pigment production, besides daptomycin production. Future studies will identify additional DasR targets involved in these biological processes.

## 4. Materials and Methods

### 4.1. Strains, Plasmids, Primer Pair, and Growth Conditions

Strains and plasmids used are listed in Appendix A, and primers are listed in Appendix A. Growth conditions for *S. roseosporus* and *Escherichia coli* strains were described previously [11,20]. Solid DA1 [11] was used for *S. roseosporus* sporulation and phenotype observation. Seed medium and fermentation medium [11] were used for daptomycin production.

### 4.2. Construction of S. roseosporus Mutant Strains

For in-frame gene deletion of *dasR*, a 427 bp 5′ flanking region (positions from −391 to +36 relative to *dasR* start codon) and a 605 bp 3′ flanking region (positions from −30 to +575 relative to *dasR* stop codon) were amplified, respectively, with primer pairs CQ9/CQ10 and CQ11/CQ12 from WT genomic DNA. The two fragments were connected by fusion PCR with primer pair CQ9/CQ12 and ligated into *Xba*I*/Hin*dⅢ-digested pKC1139 [39] to generate *dasR*-deletion plasmid pDdasR, which was then transformed into WT protoplasts. *dasR*-deletion mutant DdasR was isolated as described previously [11], confirmed by PCR with primer pairs CQ13/CQ14 (flanking exchange regions) and CQ15/CQ16 (located within deletion region of *dasR* ORF) (Appendix A), and DNA-sequenced. Use of primer pair CQ13/CQ14 generated a 1389 bp band in the mutant and a 2085 bp band in WT. When primer pair CQ15/CQ16 was used, only WT produced a 418 bp band (data not shown).

For complementation of DdasR, a 1118 bp DNA fragment carrying *dasR* ORF and its promoter was amplified with primer pair CQ75/CQ76. The PCR product was excised with *Eco*RI/*Xba*I and ligated into pSET152 [39] to generate *dasR*-complemented plasmid pCdasR, which was then transformed into DdasR to obtain complemented strain CdasR. For overexpression of *dasR*, an 897 bp DNA fragment carrying *dasR* ORF was amplified with primer pair CQ21/CQ22 and excised with *Hin*dIII/*Xba*I. A 188 bp *ermE*p* fragment was excised from pJL117 [40] with *Eco*RI/*Hin*dIII. The two fragments were ligated simultaneously into *Eco*RI/*Xba*I-digested pKC1139 to generate *dasR* overexpression plasmid pOdasR, which was transformed into WT to construct *dasR* overexpression strain OdasR.

To construct an *adpA*-gene-deletion mutant, a 423 bp 5′ flanking region (positions from −335 to +88 relative to *adpA* start codon) was amplified with primer pair CQ1/CQ2, and a 560 bp 3′ flanking region (positions from −55 to +505 relative to *adpA* stop codon) was amplified with primer pair CQ3/CQ4. The two fragments were fused by PCR with primer pair CQ1/CQ4 and ligated into *Eco*RI/*Xba*I-digested pKC1139 to generate *adpA*-deletion plasmid pDadpA, which was then transformed into WT. The mutant, termed DadpA, was isolated by selection of DdasR and confirmed by PCR with primer pairs CQ5/CQ6 (flanking exchange regions) and CQ7/CQ8 (located within deletion region of *adpA* ORF) (Appendix A). Use of primer pair CQ5/CQ6 generated a 1414 bp band in the mutant and a 2450 bp band in WT. When primer pair CQ7/CQ8 were used, only WT produced a 298 bp band (data not shown).

For complementation of DadpA, a 1705 bp DNA fragment carrying *adpA* ORF and its promoter was amplified with primer pair CQ19/CQ20, excised with *Eco*RI/*Xba*I, and ligated into pSET152 to generate *adpA*-complemented plasmid pCadpA, which was transformed into DadpA to obtain complemented strain CadpA. For *adpA* overexpression, a 1391 bp DNA fragment carrying *adpA* ORF was amplified using primer pair CQ17/CQ18, excised with *Hin*dIII/*Xba*I, and ligated simultaneously with the 188 bp *Eco*RI/*Hin*dIII-*ermE*p* fragment into *Eco*RI/*Xba*I-digested pKC1139 to generate *adpA* overexpression plasmid pOadpA, which was transformed into WT to construct *adpA* overexpression strain OadpA.

### 4.3. Production and Analysis of Daptomycin

The fermentation process of *S. roseosporus* strains and the quantitative analysis of daptomycin production by HPLC were performed as described previously [11]. Briefly, spores of *S. roseosporus* prepared from DA1 plates were added to 50 mL primary seed medium in flasks and incubated at 28 °C, 250 rpm for 48 h. The culture in primary seed medium was inoculated at 6% (*v*/*v*) into 50 mL secondary seed medium and incubated at same condition for 30 h. Then, a 6% (*v*/*v*) inoculation volume of the culture in secondary seed medium was inoculated into 50 mL fermentation medium and cultured for 10 days. Sodium decanoate (final concentration 0.02%, *w*/*v*) was added every 12 h after 48 h of fermentation until the end of fermentation.

After fermentation, the broth was centrifuged and the supernatant was analyzed by HPLC on a C_18_ reverse-phase column (5 μm, 4.6 mm × 100 mm; Waters, Milford, MA, USA) with UV detection at 218 nm at a flow rate of 1.0 mL/min. The mobile phase contained 0.1% (*v*/*v*) trifluoroacetic acid in water and acetonitrile (55:45, *v*/*v*). Authentic daptomycin sample was used as a standard.

### 4.4. Reverse Transcription and Quantitative Real-Time PCR (RT-qPCR) Analysis

Samples of *S. roseosporus* strains grown in fermentation medium were collected at various time points and frozen in liquid nitrogen. Total RNAs were prepared using TRIzol reagent (Tiangen; Beijing, China), and samples were treated with DNase I (TaKaRa; Kusatsu, Japan) to eliminate genomic DNA contamination. Reverse transcription for cDNA synthesis and quantitative PCR assay of transcription levels of tested genes using respective primer pair (Appendix A) were performed as described previously [11]. Relative transcription value of each gene was normalized relative to internal control gene *hrdB* value using comparative Ct method. Experiments were performed in triplicate.

### 4.5. Overexpression and Purification of His_6_-DasR and His_6_-AdpA

To prepare His_6_-DasR protein, an 855 bp fragment containing *dasR* ORF was amplified with primer pair CQ25/CQ26. To prepare His_6_-AdpA protein, primer pair CQ23/CQ24 was used to amplify a 1259 bp fragment containing *adpA* ORF. PCR products were excised with *Eco*RI/*Xho*I and ligated into pET-28a (+) to generate pET28-dasR and pET28-adpA, which were transformed separately into *E. coli* BL21 (DE3) for protein overexpression. Cells were induced by 0.4 mM IPTG for 10 h at 16 °C, and those containing recombinant protein were collected, washed, disrupted in lysis buffer [41] by sonication on ice, and centrifuged. Recombinant His_6_-tagged protein from supernatant was purified using Ni^2+^-NTA resin (Qiagen; Hilden, Germany), eluted with lysis buffer plus 250 mM imidazole, and dialyzed against binding buffer for electrophoretic mobility shift assays (EMSAs) to eliminate imidazole. Purified His_6_-DasR and His_6_-AdpA were stored at −80 °C until use.

### 4.6. EMSAs

Probes carrying promoter regions of tested genes were amplified using corresponding primer pair (Appendix A), labeled at the 3′ end with digoxigenin (DIG), and EMSAs were performed using a DIG gel shift kit (2nd generation; Roche, Mannheim, Germany) as described previously [41]. Each binding reaction system (20 μL) contained 1 μg poly [d(I-C)], 0.15 nM labeled probe, and various amounts of His_6_-DasR or His_6_-AdpA as indicated. Specificity of DasR (or AdpA)-probe interaction was tested by adding ~100-fold excess of each specific or nonspecific *hrdB* unlabeled probe to the reaction system.

### 4.7. DNase I Footprinting Assay

To identify binding sites of DasR and AdpA, 5′-FAM fluorescence-labeled DNA fragments corresponding to upstream regions of tested genes were PCR-synthesized using primers listed in Appendix A, and gel-purified. DNase I footprinting assays were performed using a nonradiochemical capillary electrophoresis method as described previously [11,42], and electropherogram data were analyzed using software program GeneMarker V. 2.2.0.

### 4.8. 5′ Rapid Amplification of cDNA Ends (5′ RACE)

TSSs of *dptA* and *dptR2* were identified by 5′ RACE using a 5′/3′ RACE Kit (Roche; Mannheim, Germany). A measure of 2 µg total RNA extracted from 48 h culture of *S. roseosporus* WT grown in fermentation medium was used for cDNA synthesis with 20 pmol gene-specific primer dptA-SP1 or dptR2-SP1. Obtained cDNAs were purified, and oligo-dA tails were added to 3′ ends by terminal transferase (TaKaRa). First, a PCR round was performed using tailed cDNA as template, and oligo dT-anchor primer and second inner gene-specific primer dptA-SP2 or dptR2-SP2. Second PCR round was performed to yield a specific single band, using 10-fold diluted original PCR product as template, and anchor primer and nested primer dptA-SP3 or dptR2-SP3. The final PCR product was sent for sequencing, and TSS was determined as the first nucleotide following oligo-dA sequence.

## 5. Conclusions

In summary, we characterized *S. roseosporus* DasR as a key repressor during daptomycin production through both direct and cascade mechanisms, based on binding to promoter regions of *dptE* operon and global regulatory gene *adpA*. DasR was also demonstrated to negatively regulate morphological development, but positively regulate pigment production. Furthermore, we verified AdpA function as an activator of daptomycin production and development, and identified *dptA* operon and *dptR2* as novel AdpA targets. Our work contributes to the clarification of the complex regulatory mechanisms of daptomycin production and rationally construct daptomycin overproducers.

## Figures and Tables

**Figure 1 antibiotics-11-01065-f001:**
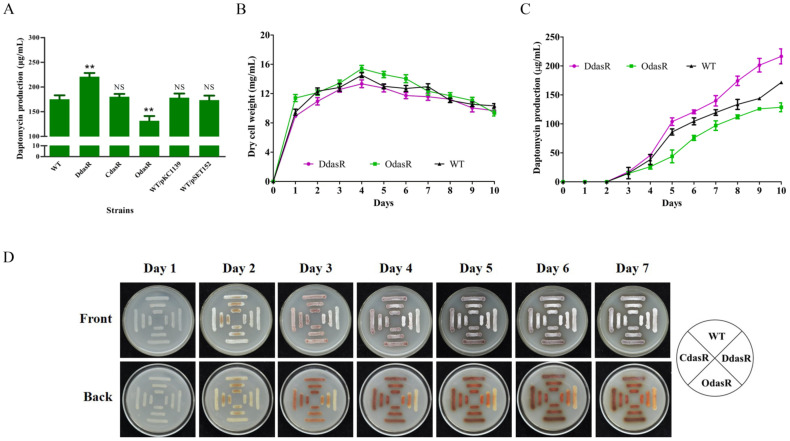
Effects of DasR on daptomycin and pigment production, cell growth, and morphological development in *S. roseosporus*. (**A**) Daptomycin production in WT, *dasR*-deletion mutant DdasR, complemented strain CdasR, overexpression strain OdasR, and plasmid control strains WT/pKC1139 and WT/pSET152 cultured in fermentation medium for 10 days. Statistical notations: NS, not significant; **, *p* < 0.01 for comparison with WT (*t*-test). (**B**) Growth curves for WT, DdasR, and OdasR. Biomass is presented as dry cell weight. (**C**) Daptomycin production curves for WT, DdasR, and OdasR. Error bars in panels (**A**–**C**): SD for three replicates. (**D**) Phenotypes of WT, DdasR, CdasR, and OdasR grown on DA1 plates at 28 °C.

**Figure 2 antibiotics-11-01065-f002:**
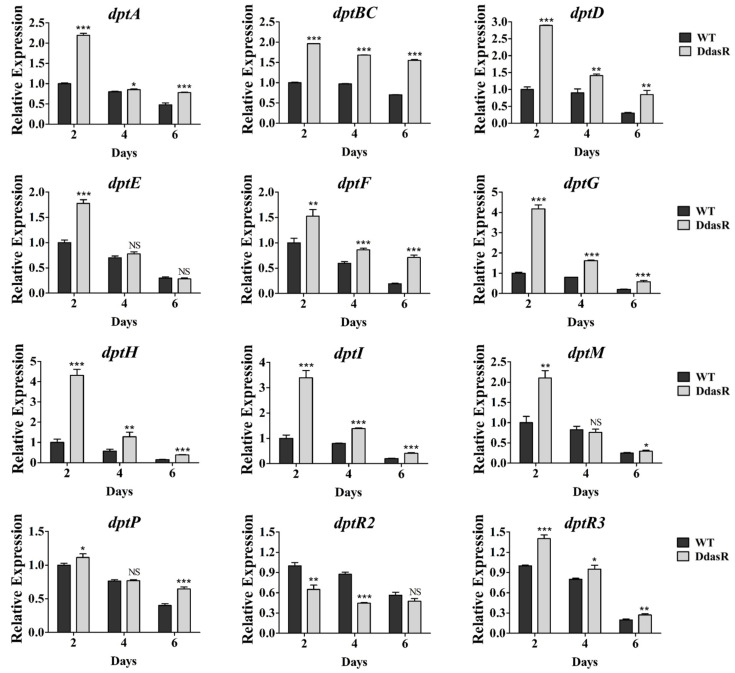
RT-qPCR analysis of *dpt* genes in WT and DdasR grown in fermentation medium for 2, 4, or 6 days. WT transcription level for each gene on day 2 was defined as 1. NS, not significant; *, *p* < 0.05; **, *p* < 0.01; ***, *p* < 0.001 (*t*-test). Error bars: SD for three replicates.

**Figure 3 antibiotics-11-01065-f003:**
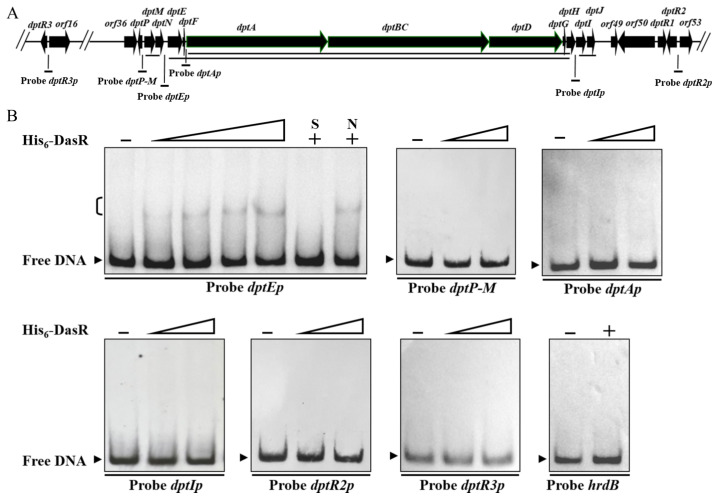
Binding of DasR to *dptE*-promoter region. (**A**) Promoter probes used for EMSAs (schematic). Long solid lines: transcriptional units. (**B**) EMSAs of interactions of His_6_-DasR with promoter probes of *dpt* genes. Negative control probe: *hrdB*. Concentrations of His_6_-DasR for probe *dptEp*: 1, 1.5, 3, and 4.5 μM; for other *dpt* promoter probes: 1 and 4.5 μM. Lanes –: EMSAs without His_6_-DasR. A measure of 4.5 μM His_6_-DasR was used for competition assays and control probe *hrdB* (lanes +). Lanes N and S: competition assays with ~100-fold excess of unlabeled nonspecific probe *hrdB* (N) and specific probe *dptEp* (S). Bracket (upper left): DasR-DNA complex. Arrowheads: free probe.

**Figure 4 antibiotics-11-01065-f004:**
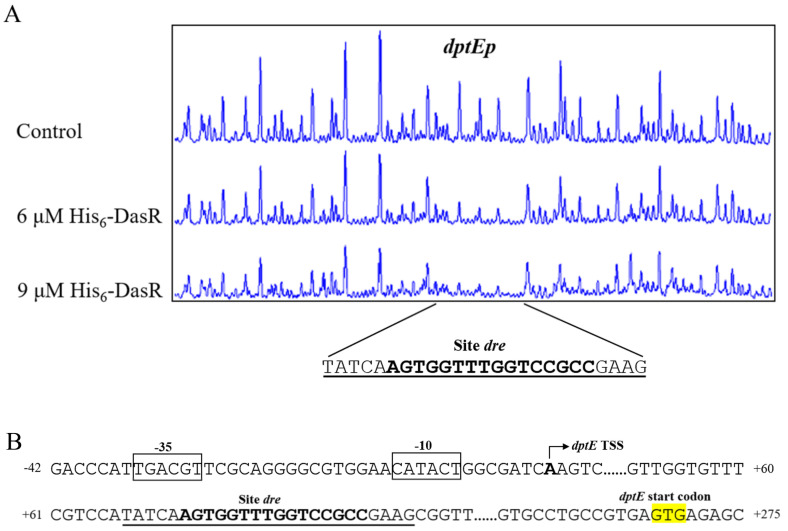
Determination of DasR-binding site on *dptEp*. (**A**) DNase I footprinting assay of DasR on *dptEp*. Protection patterns were acquired with increasing His_6_-DasR concentrations. Control: reaction without His_6_-DasR. (**B**) Nucleotide sequences of *dptE*-promoter region and DasR-binding site. Numbers: distance (nt) from *dptE* TSS. Bent arrow: *dptE* TSS. Boxes: putative −10 and −35 regions. Yellow highlight: *dptE* TSC. Solid line: DasR-binding site. Bold font: *dre*-like sequence.

**Figure 5 antibiotics-11-01065-f005:**
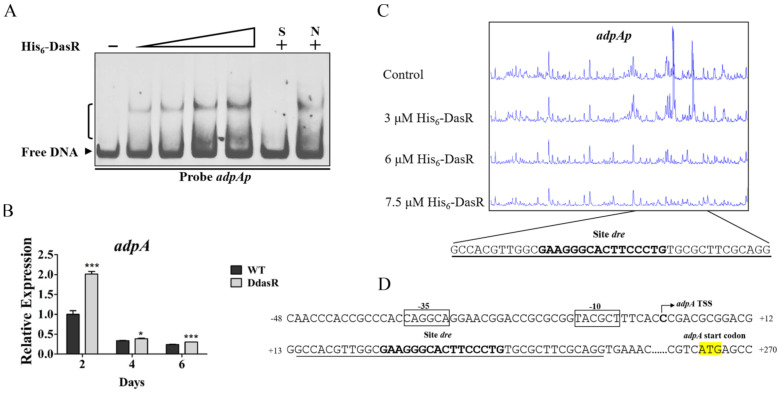
Identification of DasR target *adpA*. (**A**) EMSAs of His_6_-DasR with probe *adpAp*. Lanes 2–5: 1, 1.5, 3, and 4.5 μM His_6_-DasR. Notations are as in Figure 3B. (**B**) RT-qPCR analysis of *adpA* in WT and DdasR grown in fermentation medium. *, *p* < 0.05; ***, *p* < 0.001 (*t*-test). Error bars: SD for three replicates. (**C**) DNase I footprinting assay of DasR on *adpAp*. (**D**) Nucleotide sequences of *adpA* promoter region and DasR-binding site. Formatting conventions as in Figure 4B.

**Figure 6 antibiotics-11-01065-f006:**
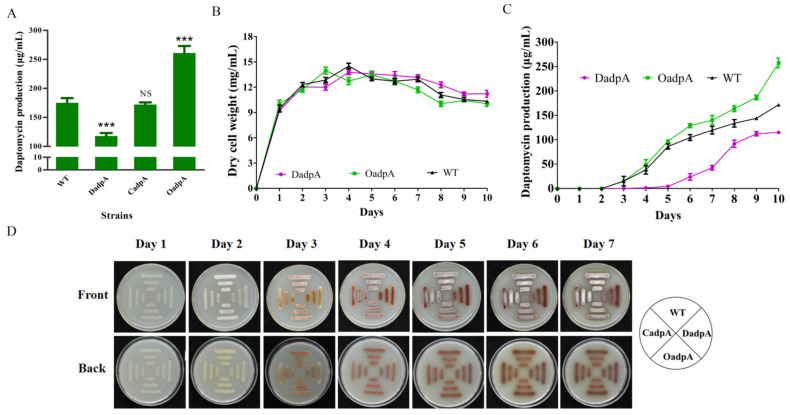
Effects of AdpA on daptomycin and pigment production, cell growth, and morphological development in *S. roseosporus*. (**A**) Daptomycin production in WT, DadpA, CadpA, and OadpA cultured in fermentation medium for 10 days. NS, not significant; ***, *p* < 0.001 (*t*-test). (**B**) Growth curves for WT, DadpA, and OadpA. (**C**) Daptomycin production curves for WT, DadpA, and OadpA. Error bars in panels (**A**–**C**): SD for three replicates. (**D**) Phenotypes of WT, DadpA, CadpA, and OadpA grown on DA1 plates at 28 °C.

**Figure 7 antibiotics-11-01065-f007:**
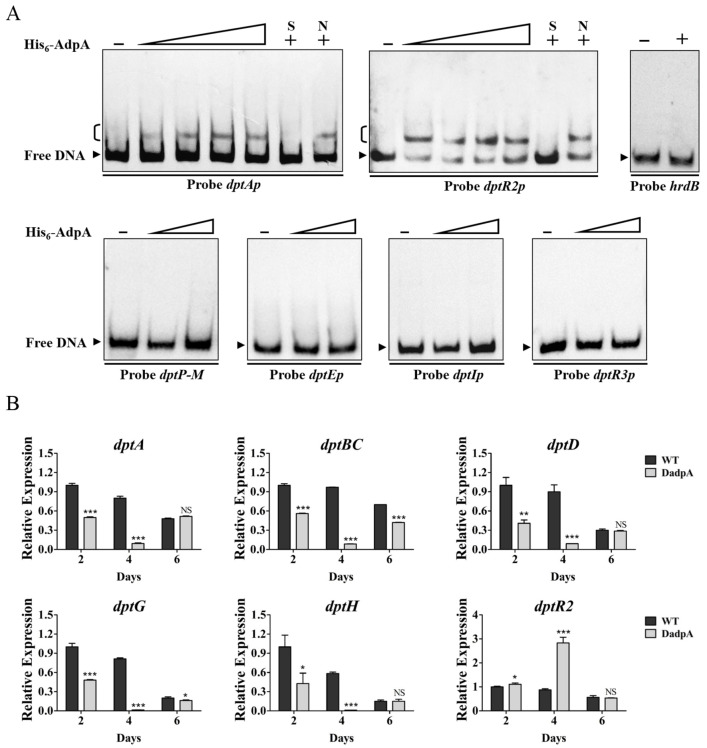
Identification of AdpA targets *dptA* and *dptR2*. (**A**) EMSAs of His_6_-AdpA with indicated promoter probes. Concentrations of His_6_-AdpA for probes *dptAp* and *dptR2p*: 50, 100, 150, and 200 nM; for control probe *hrdB*: 200 nM; for other probes: 50 and 200 nM. A measure of 200 nM His_6_-AdpA was used for competition assays. Notations are same as in Figure 3B. (**B**) RT-qPCR analysis of *dptR2* and *dpt* structural genes within *dptA* operon in WT and DadpA grown in fermentation medium. NS, not significant; *, *p* < 0.05; **, *p* < 0.01; ***, *p* < 0.001 (*t*-test). Error bars: SD for three replicates.

**Figure 8 antibiotics-11-01065-f008:**
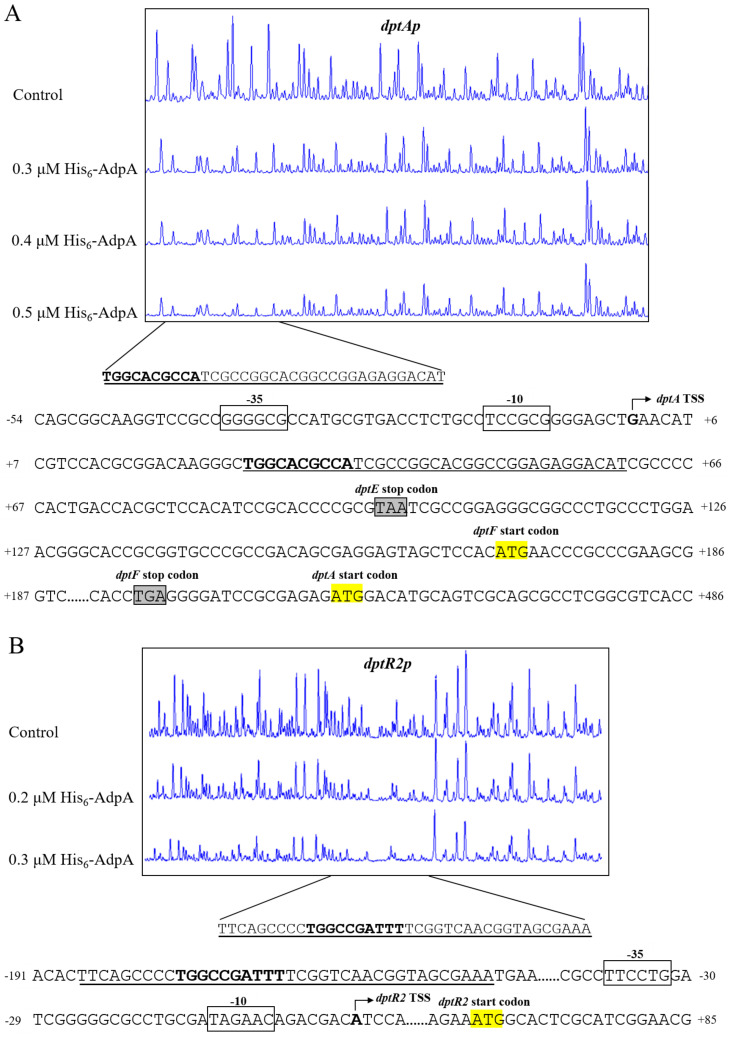
DNase I footprinting assays and nucleotide sequences of AdpA-binding sites on *dptAp* (**A**) and *dptR2p* (**B**). Numbers: distance (nt) from respective TSSs. Bent arrows: TSSs. Boxes: putative −10 and −35 regions. Yellow highlight: TSCs. Solid lines: AdpA protected regions. Bold font: sequence similar to the consensus AdpA-binding site.

**Figure 9 antibiotics-11-01065-f009:**
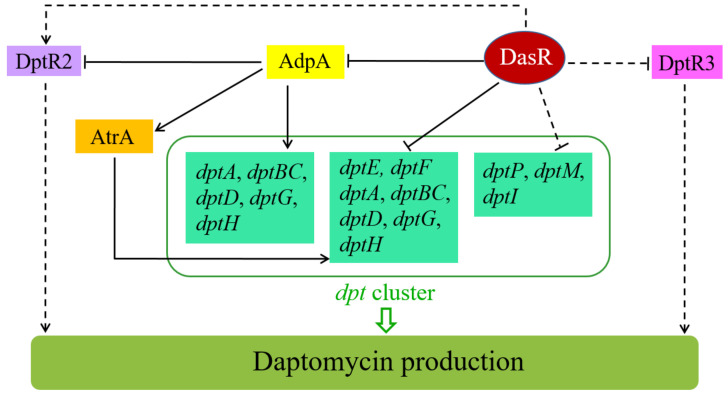
Proposed model of regulatory role of DasR in control of daptomycin production in *S. roseosporus*. Solid arrows: activation. Bars: repression. Solid lines: direct control. Dashed lines: indirect control.

## Data Availability

Not applicable.

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
