# Peer review of "Transcriptional Regulator DasR Represses Daptomycin Production through Both Direct and Cascade Mechanisms in *Streptomyces roseosporus"

_antibiotics, 2022, doi:10.3390/antibiotics11081065_

Round 1

Reviewer 1 Report

Thanks for the chance of reading the article.

I have the following comments and questions for the authors. There are many awkward phrases that I do not point out here; I only point out those where the meaning cannot be interpreted:

Please double check the article by a native English reader.

Figure 3 is not clear need to be reformat.

The paragraph 2.5 first paragraph is not clear can be rewrite in more clear format.

The conclusion need to clear and specific. My recommendation is to focus on short conclusion.

Please recheck the References order.

Author Response

Reviewer #1:

Response:

Thank you very much for your comments and suggestions.

  1. Please double check the article by a native English reader.

Response:

This manuscript has been edited by Dr. S. Anderson, a professional language editor in USA to refine the language.

  1. Figure 3 is not clear need to be reformat.

Response:

We have redone EMSA experiments and provided new Fig. 3B as you suggested.

  1. The paragraph 2.5 first paragraph is not clear can be rewrite in more clear format.

Response:

We have rewritten this paragraph as you suggested.

  1. The conclusion need to clear and specific. My recommendation is to focus on short conclusion.

Response:

We have added a short conclusion after Discussion as you suggested.

  1. Please recheck the References order.

Response:

We have added four references according to the comments and rechecked the reference order as you suggested.

Reviewer 2 Report

Chen et al. describe the partial characterisation of a complex regulatory network that controls production of the antibiotic daptomycin by S. roseosporus. In particular, the authors have focused on the regulator DasR and its direct targets. Some of these targets are shown to be transcription factors themselves, which in turn regulate downstream genes involved in daptomycin synthesis.

The manuscript is clearly written and of significant interest to the field. The experiments are well described and the figures informative.

However, a few issues should be addressed.

Fig. 3B: the amount of probe used in these experiments appears to be wildly variable, resulting in massive bands in the first panel and barely visible ones in the third (as well as anything in between in the other panels). This makes it impossible to compare binding of DasR to the various probes and draw firm conclusions from these assays. For instance, if as little dptEp had been used for the first panel as the amount of dptAp used in the third panel, then clearly the partial dptEp shift would not have been visible. Conversely, if larger amounts of dptAp had been used in the third panel, it is possible that shifts might have shown up.

The DasR concentrations required to observe appreciable binding to the regulatory elements (in the EMSAs as well as in the DNAse I footprinting experiments) are astonishingly high, in the 10-100 micromolar range. Can the authors please comment on this? Is the affinity of DasR for these operator sequences exceptionally weak compared to the ones described in the literature (e.g. reference 18)? Could DasR concentrations in the 10-100 micromolar range actually exist in S. roseosporus? Or do the authors maybe have indications that operator binding could be stronger in vivo, for instance due to cooperative interactions with other regulators? The authors might want to discuss these aspects.

Fig. 5A: the EMSA clearly shows two shifted bands, how do the authors explain these? Do the DNA sequence and/or the DNAse I footprinting experiment perhaps provide indications for a second binding site in adpAp?

Minor points:

The authors cite literature showing that both DptR2 and DptR3 promote daptomycin production. It seems counterintuitive that DasR should upregulate one while downregulating the other, as the experiments appear to indicate. Maybe the authors can comment on this?

The effect of DasR on one of the cluster-associated regulatory genes, dptR1, is not investigated at all and it is not entirely clear why this gene has been left out of the analysis. The authors only state that “the function of dptR1 is unclear” (l. 140-141). Does this mean that DptR1 does not influence daptomycin production and/or expression of the other dpt genes?

The authors state that they have used “DA1 sporulation plates” to assess effects on morphological development (l. 115-116). It would be good to provide a short explanation, as not all readers may be familiar with this approach.

In two of the bar graphs (Figs. 1A and 6A), a large part of the vertical data range (i.e. the lower part of the Y-axis) is not needed and should be omitted in order to more clearly show the differences in the measurements as well as the size of the error bars.

l. 174-175: definitions of the symbols D, S, W, etc. are given in the introduction (l. 74) and they are then repeated in l. 267-268, but not here?

Author Response

Reviewer #2:

Response:

Thank you very much for your comments and suggestions.

  1. 3B: the amount of probe used in these experiments appears to be wildly variable, resulting in massive bands in the first panel and barely visible ones in the third (as well as anything in between in the other panels). This makes it impossible to compare binding of DasR to the various probes and draw firm conclusions from these assays. For instance, if as little dptEp had been used for the first panel as the amount of dptAp used in the third panel, then clearly the partial dptEp shift would not have been visible. Conversely, if larger amounts of dptAp had been used in the third panel, it is possible that shifts might have shown up.

The DasR concentrations required to observe appreciable binding to the regulatory elements (in the EMSAs as well as in the DNAse I footprinting experiments) are astonishingly high, in the 10-100 micromolar range. Can the authors please comment on this? Is the affinity of DasR for these operator sequences exceptionally weak compared to the ones described in the literature (e.g. reference 18)? Could DasR concentrations in the 10-100 micromolar range actually exist in S. roseosporus? Or do the authors maybe have indications that operator binding could be stronger in vivo, for instance due to cooperative interactions with other regulators? The authors might want to discuss these aspects.

Response:

We repurified His6-DasR and redid EMSA experiments to ensure equal amounts of various probes (please see new Fig. 3B). The results also showed that 1, 1.5, 3, and 4.5 μM His6-DasR formed complexes with probe dptEp, but not with dptP-M, dptAp, dptIp, dptR2p, dptR3p, or hrdB at concentrations of 1.5 and 4.5 μM, confirming our previous results. In the case of DNase I footprinting experiments, we’re sorry that the concentrations of His6-DasR should be 6 and 9 μM, not 60 and 90 μM in Fig. 4A. Similarly, the concentrations of His6-DasR should be 3, 6, and 7.5 μM in Fig. 5C. We have revised Fig. 4A and 5C.

We used recombinant protein His6-DasR, which was heterologous expressed and purified from E. coli, for in vitro EMSA and DNase I footprinting experiments. The purpose of these experiments is to investigate whether DasR could bind to its target promoter regions and identify its binding sites. The concentrations of His6-DasR for in vitro EMSA and DNase I footprinting experiments could not represent the binding concentration of native DasR with its target promoters in vivo of S. roseosporus. The relatively low DNA affinity of His6-tagged DasR might be due to abnormal folding of DasR caused by His6 tag or by heterologous host cell E. coli.

  1. 5A: the EMSA clearly shows two shifted bands, how do the authors explain these? Do the DNA sequence and/or the DNase I footprinting experiment perhaps provide indications for a second binding site in adpAp?

Response:

We repurified His6-DasR and EMSA results showed that fresh His6-DasR also generated two shifted bands on adpAp when protein concentration reached 3 μM (please see new Fig. 5A). According to our DNase I footprinting result, DasR only has one binding site on adpAp. The number of shifted bands is not always corresponding to the number of binding sites. Sometimes one binding site could generate two or more shifted bands, and this phenomenon might be due to the different polymerization forms of protein or conformational change upon binding of protein to the target DNA. In the case of BldD, it was also reported that S. coelicolor BldD bound one site to its own promoter region (J Bacteriol, 1999,181:6832-5) and to bdtA promoter region (Mol Microbiol, 2001, 40:257-269), but generated two shifted bands in EMSAs, consistent with our previous results that BldD generated two shifted bands with bldDp, dptEp and dptR3p, but only one BldD-binding site was revealed on these promoter regions (Mol Microbiol, 2020, 113:123-142). DasR binds DNA in the dimeric form and its binding site is a 16-bp imperfect palindromic sequence. One possibility is that different forms of DasR dimer generate different shifted bands. Another possibility is that the first shift represents binding to the higher-affinity half site, and then a conformational change permits binding to the other half site, which is seen as the second shift.

  1. The authors cite literature showing that both DptR2 and DptR3 promote daptomycin production. It seems counterintuitive that DasR should upregulate one while downregulating the other, as the experiments appear to indicate. Maybe the authors can comment on this?

Response:

We repeated qRT-PCR analysis several times and the results all showed that dptR2 was downregulated whereas dptR3 was upregulated in DdasR. We think the results are not contradictory. Because the regulatory effect of DasR on dptR2 and dptR3 is indirect but not direct. Based on our findings in this work, DasR indirectly activates dptR2 expression through AdpA, i.e., it directly represses adpA, whose product AdpA directly represses dptR2 (see Fig. 9), which leads to downregulated dptR2 in DdasR. Although we don’t know the mechanism whereby DasR indirectly represses dptR3, it does not through AdpA, based on our finding that dptR3 is not AdpA target. Therefore, DasR exerts its regulatory effect on dptR2 and dptR3 through different mechanisms, and it is not surprising that dptR2 was downregulated whereas dptR3 was upregulated in DdasR. The observed change of daptomycin production in DdasR reflects a combined effect of altered transcription levels of dpt structural genes and regulatory genes dptR2 and dptR3.

  1. The effect of DasR on one of the cluster-associated regulatory genes, dptR1, is not investigated at all and it is not entirely clear why this gene has been left out of the analysis. The authors only state that “the function of dptR1 is unclear” (l. 140-141). Does this mean that DptR1 does not influence daptomycin production and/or expression of the other dpt genes?

Response:

Because we found that DptR1 did not affect daptomycin production, we did not investigate the effect of dasR deletion on dptR1 expression. We have rewritten this sentence for better understanding. 

  1. The authors state that they have used “DA1 sporulation plates” to assess effects on morphological development (l. 115-116). It would be good to provide a short explanation, as not all readers may be familiar with this approach.

Response:

We have changed “on DA1 sporulation plates” to “on DA1 plates for sporulation” and also explained in section 5.1 that solid DA1 was used for S. roseosporus sporulation and phenotype observation (please see L402) and in the Introduction section that the Streptomyces morphological development involves formation of substrate hyphae, aerial hyphae, and chains of spores (please see L35-36).

  1. In two of the bar graphs (Figs. 1A and 6A), a large part of the vertical data range (i.e. the lower part of the Y-axis) is not needed and should be omitted in order to more clearly show the differences in the measurements as well as the size of the error bars.

Response:

We have changed the vertical data range of Figs. 1A and 6A as you suggested.

  1. Lines 174-175: definitions of the symbols D, S, W, etc. are given in the introduction (l. 74) and they are then repeated in l. 267-268, but not here?

Response:

We have added definitions of the symbols (D = A/G/T; S = C/G; W = A/T; V = A/C/G; M = A/C; H = A/C/T; B = G/C/T; N = A/G/C/T) here as you suggested.

Reviewer 3 Report

The paper demonstrates the function of the DasR transcriptional regulator in Streptomyces roseosporus. It has been shown that DasR binds the daptomycin gene cluster by the interaction with the promoter of dptE from the core operone. Furthermore, DasR interacts with the promoter of the gene adpA encoding another transcriptional regulator AdpA that targets dpt structural genes and further regulatory genes. Remarkably, DasR acts as a repressor – the deletion of the dasR gene lead to the increase of trascription of daptomycin structural genes and other regulatory genes as well as to the increase of daptomycin production. Thus, this regulator can influence daptomycin production directly through targeting the daptomycin gene cluster and indirectly targeting the genes that encode regulators of daptomycin production (cascade mechanism). These findings can be potentially applied in biotechnology for optimization of daptomycin production. The paper is technically sound (a variety of convincing methods was used: Streptomyces growth and phenotypical analysis of strains with the deletion, complementation and overexpression of dasR and adpA; RT-qPCR; EMSA assays; DNase I footprinting assays; HPLC for daptomycin detection). The claims are convincing – they are supported by the experimental data and statistical analysis. Overall, this is a solid contribution to the Streptomyces and antibiotics research field with consideration of some suggestions for improvement:

1. Lines 47-49: I suggest to add a short description of the structure / biosynthesis of daptomycin, as well as clarification of dptE-dptF-dptA-dptBC-dptD-dptG-dptH and dptI, dptJ gene products?

2. Lines 122, 243: Error bars B in Fig 1 and 6 (biomass measurements) are not visible

3. Lines 319-320: For a better comprehension by the readers - are there any concrete differences in the core genome of the S. roseosporus strains used in this study (NRRL11379) and in the previous study? Some other papers report results also for “industrial strains” like SR1101. Was the strain for this study developed in the same group or derived from another one?

4. Line 325: The global regulator Crp can be addressed. In some previous reports it has been demonstrated that the Crp regulator has a positive effect on daptomycin biosynthesis in S. roseosporus (https://doi.org/10.3389/fbioe.2021.618029). Crp has been also characterized in S. coelicolor (https://journals.asm.org/doi/full/10.1128/mBio.00407-12). Another question: may there be influence from the primary metabolism (flux of precursors for daptomycin synthesis) in S. roseosporus?

5. Line 357: There is a double space before “Future”

6. Line 373: in HindIII “d” not italics

Author Response

Reviewer #3:

Response:

Thank you very much for your comments and suggestions.

  1. Lines 47-49: I suggest to add a short description of the structure/biosynthesis of daptomycin, as well as clarification of dptE-dptF-dptA-dptBC-dptD-dptG-dptH and dptI, dptJ gene products?

Response:

We have added the description of the structure/biosynthesis of daptomycin, as well as clarification of dpt gene products as you suggested.

  1. Lines 122, 243: Error bars B in Fig 1 and 6 (biomass measurements) are not visible.

Response:

We have added error bars in Figs. 1B and 6B.

  1. Lines 319-320: For a better comprehension by the readers - are there any concrete differences in the core genome of the roseosporus strains used in this study (NRRL11379) and in the previous study? Some other papers report results also for “industrial strains” like SR1101. Was the strain for this study developed in the same group or derived from another one?

Response:

  1. roseosporus NRRL11379 we used in this study is a standard wild-type strain. This strain was kindly provided by Professor Yinghua Lu (Xiamen University, Xiamen, China). Based on Yinghua Lu’s previous report (Bioprocess Biosyst Eng, 2014, 37:415–423), strain NRRL11379 was purchased from NRRL collection. We have added this reference for the source of strain NRRL11379 (L117, L343, ref26) and thank Professor Yinghua Lu for providing us NRRL11379 in Acknowledgments.

We don’t know the source of S. roseosporus SW0702 used by YQ Li’s group. In their previous reports, sometimes they called SW0702 as wild-type strain, sometimes as industrial strain. The industrial strains such as L30 et al. they used derived from SW0702. We also don’t know the concrete differences in the core genome of the S. roseosporus strains NRRL11379 and SW0702. Thus, we just speculated that the different parent strains might lead to different results: the adpA deletion mutant constructed based on S. roseosporus SW0702 abolished daptomycin production, whereas the mutant constructed based on strain NRRL11379 in the present study reduced -- but did not abolish -- daptomycin production. On the other hand, the media we used for growth of S. roseosporus strains and daptomycin production are different from those used by YQ Li’s group, which might also contribute to different results.

  1. Line 325: The global regulator Crp can be addressed. In some previous reports it has been demonstrated that the Crp regulator has a positive effect on daptomycin biosynthesis in S. roseosporus (https://doi.org/10.3389/fbioe.2021.618029). Crp has been also characterized in S. coelicolor (https://journals.asm.org/doi/full/10.1128/mBio.00407-12). Another question: may there be influence from the primary metabolism (flux of precursors for daptomycin synthesis) in S. roseosporus?

Response:

In this paragraph, we only discussed regulators that can bind to dptEp, i.e., directly regulate dptE operon. Although Crp has a positive effect on daptomycin biosynthesis, and transcriptome and qPCR analyses showed that it can affect transcription levels of dptE and dptF (https://doi.org/10.3389/fbioe.2021.618029), there is no evidence that Crp can bind to dptEp, i.e., whether the regulatory effect of Crp on daptomycin biosynthesis is direct or indirect is unclear. Therefore, we did not mention Crp here. However, we added Crp function in daptomycin biosynthesis in the Introduction section (please see L72-74).

RNA-seq analysis showed that Crp modulates primary metabolism and enhances precursor flux to secondary metabolite biosynthesis (https://doi.org/10.3389/fbioe.2021.618029). However, as we mentioned above, we only discussed regulators that directly regulate dptE operon in this paragraph and we don’t know whether Crp can bind to dptEp; therefore, we did not discuss Crp function here.

  1. Line 357: There is a double space before “Future”.

Response:

We have deleted a space before “Future”.

  1. Line 373: in HindIII “d” not italics.

Response:

We have changed “HindIII” to “HindIII”, i.e., “d” is not italic.

Reviewer 4 Report

The study by Chen et al. identified DasR as a multifunctional regulator in Streptomyces roseosporus. The results presented in their study that DasR represses the production of daptomycin and morphological development, whereas it promotes pigment production. The overall design and experiments seem robust and convincible partially and should be acceptable with some major modifications and additional experiments and explanations.

1) Although the applied sense of this study is obvious, this reviewer is not very positive and clear with the detailed molecular mechanisms of the story. Thus the authors should reframe the manuscript based on the findings in "overproduction n" of daptomycin.

2) If the authors want to resolve the fundamental mechanism involved in their observations, the first suggestion is to take advantage of RNA-seq or Chip-seq. The data presented in this manuscript only focus on the core dpt operon which the detailed picture of DasR is still elusive. Besides, the EMSA results from Fig. 3 seem problematic regarding the banding between DasR and dptR3p probe. It is also strange to use a lower amount of other probes as compared to dptEp. This dataset (and Fig 7A) is invalid and thus MUST be changed.

3) This reviewer also suggests some morphological cell images of all mutant strains be provided as main figures. Is there some cell shape or size changes after DasR deletion? 

4) Some essential protocols and methods should be provided such as section 4.3.

Author Response

Reviewer #4:

Response:

Thank you very much for your comments and suggestions.

  1. Although the applied sense of this study is obvious, this reviewer is not very positive and clear with the detailed molecular mechanisms of the story. Thus the authors should reframe the manuscript based on the findings in "overproduction" of daptomycin.

Response:

We have reframed the manuscript in "overproduction" of daptomycin as you suggested.

  1. If the authors want to resolve the fundamental mechanism involved in their observations, the first suggestion is to take advantage of RNA-seq or ChIP-seq. The data presented in this manuscript only focus on the core dpt operon which the detailed picture of DasR is still elusive. Besides, the EMSA results from 3 seem problematic regarding the banding between DasR and dptR3p probe. It is also strange to use a lower amount of other probes as compared to dptEp. This dataset (and Fig 7A) is invalid and thus MUST be changed.

Response:

The main purpose of this work is to investigate whether DasR could directly regulate dpt cluster for daptomycin production. So we did not perform RNA-seq or ChIP-seq. We demonstrated that DasR acts as a key repressor during daptomycin production by binding to promoter regions of the core dptE operon within dpt cluster and also global regulatory gene adpA. Thus, DasR regulates daptomycin production both directly and in a cascade manner.

We repurified His6-DasR and His6-AdpA proteins and redid all of the EMSA experiments to ensure equal amounts of various probes as you suggested. Please see new Fig. 3B, Fig. 5A, and Fig. 7A.

  1. This reviewer also suggests some morphological cell images of all mutant strains be provided as main figures. Is there some cell shape or size changes after DasR deletion?

Response:

As shown in Fig. 1D, deletion of dasR (strain DdasR) just caused earlier sporulation, but did not lose the capability of producing aerial hyphae and spores. When prolonged incubation (≥6 days), the mutant DdasR exhibited similar phenotype to WT strain. Therefore, we did not show cell images of mutant DdasR. In the case of AdpA, YQ Li’s group had reported that deletion of adpA caused bald phenotype, i.e., grew only in substrate hyphae (J Biol Chem. 2015, 290:7992-8001). In this work, we just confirmed AdpA function in development. As shown in Fig. 6D, adpA deletion mutant DadpA showed bald phenotype, consistent with the study by YQ Li’s group. Therefore, we also did not show cell images of mutant DadpA.

  1. Some essential protocols and methods should be provided such as section 4.3.

Response:

We have provided protocols for section 4.3 (new 5.3) and more information for sections 4.1 (new 5.1), 4.6 (new 5.6), and 4.7 (new 5.7) as you suggested.

Round 2

Reviewer 1 Report

Accept în present format!

Reviewer 4 Report

This reviewer has no further comments on this revised manuscript.